# Recent Progress on Techniques in the Detection of Aflatoxin B_1_ in Edible Oil: A Mini Review

**DOI:** 10.3390/molecules27196141

**Published:** 2022-09-20

**Authors:** Shipeng Yin, Liqiong Niu, Yuanfa Liu

**Affiliations:** 1School of Food Science and Technology, Jiangnan University, No. 1800 Lihu Road, Binhu District, Wuxi 214122, China; 2School of Life Sciences, Guangzhou University, Guangzhou 510006, China

**Keywords:** aflatoxin B_1_, edible oil, chromatographic technology, spectroscopic technology, biosensor technology, recognition elements

## Abstract

Contamination of agricultural products and foods by aflatoxin B_1_ (AFB_1_) is becoming a serious global problem, and the presence of AFB_1_ in edible oil is frequent and has become inevitable, especially in underdeveloped countries and regions. As AFB_1_ results from a possible degradation of aflatoxins and the interaction of the resulting toxic compound with food components, it could cause chronic disease or severe cancers, increasing morbidity and mortality. Therefore, rapid and reliable detection methods are essential for checking AFB_1_ occurrence in foodstuffs to ensure food safety. Recently, new biosensor technologies have become a research hotspot due to their characteristics of speed and accuracy. This review describes various technologies such as chromatographic and spectroscopic techniques, ELISA techniques, and biosensing techniques, along with their advantages and weaknesses, for AFB_1_ control in edible oil and provides new insight into AFB_1_ detection for future work. Although compared with other technologies, biosensor technology involves the cross integration of multiple technologies, such as spectral technology and new nano materials, and has great potential, some challenges regarding their stability, cost, etc., need further studies.

## 1. Introduction

Food security has always been an issue of concern in the international community, and, in recent years, food contamination has become a major factor affecting food security. Contaminated food can not only adversely influence human health (poisoning events, chronic diseases, etc.) but also affect and slow down the economy. When people consume contaminated food, they need to spend a lot of money and time on treatment. There are many factors causing food contamination, such as biological, chemical, and physical factors. Among these, microbial contamination is common and mainly includes contamination by bacteria, fungi, molds, viruses, or their toxins and by-products [1,2]. Mycotoxins are common food contaminants, which can cause changes in the appearance, flavor, smell, and other characteristics of food [3,4,5,6,7]. Mycotoxins are secondary metabolites produced by fungi (e.g., *Fusarium*, *Aspergillus*, and *Penicillium*) that have multiple toxic effects on organisms and contaminate agricultural products (cereals, milk, etc.). More than 400 kinds of mycotoxins have been identified. Among them, aflatoxins (AFs) have become one of the major concerns due to their high toxicity and carcinogenicity, causing approximately 25% of animal deaths [8,9,10,11,12].

Edible vegetable oil plays an irreplaceable role in the human diet. The world oil crop output has increased year by year and had reached 635.5 million tons by 2021 [13]. From the growth of oil crops to the final product, i.e., oil, each link may be affected by external factors (such as mycotoxins), which may affect the quality and safety of edible vegetable oil [14]. This is because most oil crops, such as corn, peanut, soybean, rapeseed, sunflower seeds, olives, and nuts, are seasonal. During the growth process, they will be affected by climate, pests, and other factors and can be easily be infected by *Aspergillus* flavus. After harvest, the oil may deteriorate or be affected by mildew due to storage conditions (such as temperature and humidity, etc.) and storage methods [15]. At the same time, during the production of edible oil, fresh-pressed edible oil is vulnerable to contamination of raw materials infected with *Aspergillus* by aflatoxin B_1_ (AFB_1_) [16,17,18,19,20,21,22]. Therefore, contamination of edible vegetable oil products by AFB_1_ is a serious food safety problem (Figure 1) [20,23,24,25].

The presence of aflatoxin is usually detected by using precision instruments, such as high-performance liquid chromatography–mass spectrometry (HPLC–MS), high-performance liquid chromatography–fluorescence detection (HPLC–FD), or other molecular techniques, while rapid detection is mainly realized by enzymatic immunoassay ELISA [26,27]. Although different methods are available for the detection of AFB_1_ toxicity, these methods require expensive equipment and complex sample pretreatment or can only be performed at relatively high concentrations [28]. Therefore, simple, sensitive, efficient, economical, rapid, and stable AFB_1_ detection methods are required. Recently, new technologies, such as biosensors, have been applied in many fields, such as health care and food detection. Because of their key advantages, such as convenient operation, rapid response, and excellent portability, these technologies can detect harmful substances in food sensitively and accurately, helping effectively avoid their harmful effects. They have attracted increasing attention of researchers and also promoted the rapid development of biosensors. With progress in nanotechnology, scientists are paying special attention to biosensors based on nanomaterials. These new biosensors or detection systems are sensitive, rapid, consistent, and cost-effective and can be used to detect AFB_1_ in food [29,30,31,32,33].

Regarding the increased importance of biosensors for accurate detection of AFB_1_ in edible oil, we have summarized the recent advances in biosensors for AFB_1_ analysis, specifically from the points of view of the development of novel bioinspired recognition elements and nanomaterials-based electrochemical biosensors.

Therefore, we searched PubMed and web of science for publications describing the detection technology of aflatoxin B1 in edible oil. Search terms were as follows: aflatoxin B_1_ OR AFB_1_ OR *Aspergillus* OR mycotoxins OR AFB_2_ OR AFG_1_ OR AFG_2_ OR AFM_1_ OR AFM_2_ OR AFs OR AFBO OR CYP450 OR edible oil OR vegetable oil OR corn oil OR peanut oil OR soybean oil OR sesame oil OR rapeseed oil OR sunflower seeds oil OR olives oil OR nuts oil OR maize oil OR canola oil OR blend oil OR coconut oil OR almond oil OR rice oil OR palm oil OR tea oil OR chromatographic technology OR spectroscopic technology OR immunological technology OR biosensor technology OR QuEChERS OR Fluorescence spectrophotometry OR Infrared spectroscopy OR Terahertz spectroscopy OR surface-enhanced raman spectroscopy (SERS) OR enzyme-linked immunosorbent assay (ELISA) OR amperometric OR impedometric OR electrochemical impedance spectroscopy (EIS) OR voltammetry (potentiometric) OR Conductometric OR LOD OR chromogenic OR Luminogenic OR Chemiluminescence OR Gravimetric OR Piezoelectric OR Magnetoelastic OR Acoustic OR electrodes (SPEs)OR SRP OR biosensors OR Nanomaterial-based biosensors OR electrochemical biosensors OR bioinspired recognition elements OR antibodies OR aptamers OR molecularly imprinted polymers OR Phylogenetic Evolution of Ligands for Exponential Enrichment (SELEX) OR fluorescence resonance energy transfer (FRET).

Publications until 29 August 2022 were included. This review only had the detection technology targeted at aflatoxin B1 in edible oil, and that had not included other types of toxins or other food carriers. After 4692 publications were searched, 596 full-text articles were reviewed and 132 articles were finally identified to meet our requirements.

## 2. Importance of Aflatoxins

Aflatoxins are a type of mycotoxins. They are highly toxic metabolites of fungi, produced in food and agricultural products. They have severe toxic effects, such as immunosuppressive, nephrotoxic, teratogenic, carcinogenic, and mutagenic, on human and animal health [34,35,36,37,38].

Aflatoxins can be divided into aflatoxin B_1_ (AFB_1_), aflatoxin B_2_ (AFB_2_), aflatoxin G_1_ (AFG_1_), and aflatoxin G_2_ (AFG_2_) according to their fluorescence properties and chromatographic mobility (Figure 1) [39,40,41]. Aflatoxin M_1_ (AFM_1_) and aflatoxin M_2_ (AFM_2_) are hydroxylated metabolites of AFB_1_ and AFB_2_, respectively. AFB_1_ is the most toxic among all AF species, with a high incidence rate and the most complex detection mechanism (Figure 2) [42].

AFB_1_ is a powerful carcinogenic, teratogenic, mutagenic, immunotoxic, hepatotoxic, and reproducible poison. Previous studies have shown that the toxicity of AFB_1_ is 10, 68, and 416 times that of KCN, arsenic and melamine, respectively [43,44] (Figure 2). Therefore, AFB_1_ has been classified as a class 1 carcinogen by many international authoritative organizations or institutions [45,46]. Due to the structural double bonds in the furan ring, AFB_1_ has high carcinogenicity and toxicity [17,47]. The lipophilic structure of atrial fibrillation promotes its entry into the blood through gastrointestinal and respiratory tracts [48,49]. Once AFB_1_ enters blood, it is distributed in various tissues and accumulates in the liver or other organs, resulting in liver cancer (Figure 3). In the liver, AFB_1_ produces a variety of metabolites through the hydroxylation and demethylation of the first-stage drug metabolism enzymes (for example, cytochrome P450 oxidase and CYP450 superfamily members, such as CYP1A2, CYP3A4, and CYP2A6) [50]. Metabolic reaction (internal and external) activates the final carcinogen AFB_1_ -8,9-epoxy metabolite, which covalently binds to cellular macromolecules (DNA, RNA, or protein) and plays a key role in acute and chronic poisoning. AFB_1_ residues also destroy the function of tumor suppressor genes (p53 and Rb) in the liver, which affects normal cells and leads to liver injury, increasing the probability of tumor and liver cirrhosis [51,52,53,54,55]. It is estimated that about 30% of liver cancers in the world are caused by AFB_1_. Its toxicity increases the infection rate of hepatitis B virus (HBV) and the risk of liver cancer [56]. A recent study found that the synergistic effect of AFB_1_ and HBV leads to liver cancer [50]. The reason is that HBV infection directly or indirectly exposes hepatocytes to AFB_1_ sensitive to tumors. The toxic effect of AFB_1_ is also related to dose, age, sex, nutrition, exposure time, and type [57]. In addition, AFB_1_ can be transmitted to the fetus through the placenta and affect the health of infants [58]. AFB_1_ exposure also inhibits immunity, thereby increasing the susceptibility to immunodeficiency virus attack and the probability of infection with other infectious diseases [59,60,61,62,63].

## 3. AFB_1_ Regulations on Edible Oil

Because AFB_1_ poses many hazards to the human body, many governments and international research institutions have made many efforts to control AFB_1_ pollution in different foods. For example, the FAO and the European Commission and Codex Alimentarius Commission have formulated regulations regarding the content of AFB_1_ in various foods to ensure consumer safety [64,65,66,67,68,69].

As for edible oil, most countries have no legislative restrictions and only a few countries, such as China, have effective regulations, laws, and standards for the highest level of AFB_1_ in different edible oils (Table 1). Due to some adverse conditions in the traditional oil processing process, AFB_1_ is usually degraded to the normal level in the extraction and refining process [17,70]. The EU has strict regulatory norms. The total amount of AFB_1_ and AF allowed in oilseeds is restricted to 2 and 4 μg kg^−1^, respectively. However, the maximum limit of AFs in oils has not been determined. The corresponding regulations in China, the United States, Kenya, and Thailand clearly stipulate the maximum level of total AFs in all edible oils, but the maximum level required is different. It is worth mentioning that in China, the AFB_1_ limit in corn and peanut oil is stipulated to be 20 μg kg^−1^, which may be because corn and peanut are most vulnerable to aflatoxin pollution [71,72]. See Table 1 for specific differences.

## 4. Methods for Detecting AFB_1_ in Edible Oil

The matrix is too complex for edible oil, and the mycotoxin content is relatively low, making it difficult to detect AFB_1_. Therefore, researchers have developed various traditional and modern methods to detect AFB_1_ in oil. AFB_1_ detection technology is mainly divided into chromatographic technology, spectroscopic technology, immunological technology, and biosensor technology [16,77].

Figure 4 briefly summarizes the LOD timelines for AFB_1_ detection in edible oils published from 2007 to 2022 mentioned in this review. It can be seen from the figure that with the advancement of time, no matter what type of detection technology or which specific detection method is used, the LOD of AFB_1_ in edible oil tends to be lower. This shows that people have a great interest in the detection of AFB1 in edible oil. At the same time, the wide use of new materials represented by nanomaterials highlights the interdisciplinary characteristics of new sensors. Next is a brief introduction of the identification method of AFB_1_, including its advantages and disadvantages, combined with actual cases.

### 4.1. Chromatographic Technology

#### 4.1.1. High Performance Liquid Chromatography (HPLC)

High-performance liquid chromatography is a common official detection method. Many countries and institutions have used it, such as China’s national standard, the European Committee for Standardization (CEN), and the association of analytical organizations (AOAC). One characteristic of the HPLC method is that it can measure multiple targets with high sensitivity [78]. In recent years, researchers have developed new detection strategies combining HPLC with other sensors, such as fluorescence detection (FLD), ultraviolet (UV) detection, diode array detection, and mass spectrometry (MS) [79,80]. Compared to traditional HPLC, this further improves the reliability, sensitivity, and accuracy of target analytes and is widely used to detect harmful substances in food. For example, HPLC combined with FLD is the standard method for detecting AFB_1_ in edible vegetable oil [81,82,83,84,85,86]. HPLC–FLD was able to detect AFB_1_ levels as low as 0.01–0.04 μg kg^−1^ [81] and 0.005–0.03 μg L^−1^ [82].

Recently, liquid chromatography–tandem mass spectrometry (LC–MS–MS) methods are being increasingly used for the analysis of mycotoxins [85]. They have the advantages of not having a sample purification limitation during extraction, high resolution, high sensitivity, and suitability for various edible vegetable oils [19,87,88,89,90,91,92,93,94,95,96,97,98,99]. GC analysis is mostly used for volatile substances, and most mycotoxins are non-volatile, further limiting the application of GC in mycotoxin detection. A similar procedure to HPLC, UHPLC or UPLC is also used on the column to improve the resolution of AFB_1_. Hidalgo et al. [100] developed a new analytical method by coupling UHPLC to a triple quadrupole analyzer (UHPLC–QqQ–MS/MS), which was well validated and applied to monitor mycotoxins, including AFB_1_, in 194 samples of edible vegetable oil.

Many commonly used methods require sample preparation due to the different matrices of edible oil. Currently, a variety of methods are available for the extraction and isolation of mycotoxins from oil, such as liquid–liquid extraction or partitioning (LLE), frequently reported in the literature [101,102,103,104,105]; solid–phase extraction (SPE) [105,106,107,108,109]; immune affinity columns (IACs) [81,94]; IAC combined with dispersive liquid–liquid microextraction (DLLME) [91]; multifunctional cleanup columns [110]; the QuEChERS system [90]; gel permeation chromatography (GPC) [111]; immune assay extraction; and low-temperature cleanup (LTC) [112,113,114,115]. However, each method has its advantages and limitations. Thus, which method to choose still depends on the type of food matrix, mycotoxin characterization, and detection techniques [116].

#### 4.1.2. Thin-Layer Chromatography (TLC)

Thin-layer chromatography (TLC) is an adsorption thin-layer chromatographic separation method suitable for complex mixed samples [117,118]. Since its development in the 1950s, thin-layer chromatography has been widely used in, for example, biology, medicine, and the chemical industry. It has recently been used in food analysis and quality control and has become a conventional technology in laboratories. Many reports have shown that TLC can be applied to all stages of the food industry, such as the stage of traditional substances, represented by food raw materials, ingredients, and additives, and the stage of unconventional substances, represented by harmful substances and pollutants. The detection and determination of compounds cover almost all substance categories [119,120,121,122].

Thin-layer chromatography uses the different adsorption capacities of each component to the same adsorbent so that when the mobile phase (the solvent) is flowing through the stationary phase (the adsorbent), there is continuous adsorption, desorption, readsorption, and redesorption to achieve the mutual separation of each component [123].

Although the TLC method has matured, it still has shortcomings, such as a low detection accuracy, volatility during the experiment being harmful to the experimental operators and the environment, and complex sample pretreatment [124,125]. In recent years, an interdisciplinary approach, such as the combination of TLC with image analysis and with new technologies, such as surface-enhanced Raman spectroscopy, mass spectrometry, and nuclear magnetic resonance, has further promoted the development of thin-layer chromatography and enhanced the practicability of this method in food analysis [126,127,128,129]. TLC is used to detect harmful substances in various foods, such as AF in edible oil, making it an effective analytical tool in food science methods [124,130].

### 4.2. Spectroscopic Technology

#### 4.2.1. Fluorescence Spectrophotometry

Spectrum-based sensing technology has been developed and used to assess AFs contamination in food [131]. Among many spectral techniques, fluorescence spectrometry shows certain potential in determining AFs in a variety of agricultural products and foods [125,132,133]. Fluorescence spectrometry uses the target molecules in the sample to absorb ultraviolet or visible light to produce fluorescence and determine its molecular structure. It has excellent detection sensitivity and specificity in the study of AFs and other chemical components [134,135]. The study found that the fluorescence phenomenon is conducive to the characterization and monitoring of target detection objects. For example, AFB_1_ can emit a specific range of fluorescence (425–500 nm) under the excitation of UV light source (340–400 nm), which provides the possibility of using fluorescence spectroscopy to analyze AFB_1_ in different foods [135,136]. In recent years, laser-induced fluorescence (LIF) technology has developed rapidly and attracted more attention because it uses a certain wavelength of laser light source and has better specificity and detection sensitivity. The advantage of LIF is that it can realize online, rapid and nondestructive direct detection according to the characteristic fluorescence peak of AFB_1_. Researchers have developed a detection model based on LIF, which can quickly and accurately screen AFB_1_ in different edible oils. The information and conclusions obtained in the study further show that LIF technology can be used for rapid and nondestructive detection of AFB1 in different edible oils [19,137]. However, LIF is also vulnerable to the interference of external factors, such as the power and accuracy of the instrument, the environmental factors of temperature and humidity, and the physical and chemical index factors of the detected object. Although this limits the wide application of LIF technology, researchers are still trying and exploring.

#### 4.2.2. Infrared (IR) Spectroscopy

Infrared spectroscopy (IRs) has the characteristics of rapid detection, simple sample preparation process and strong adaptability. It has been widely proven to be an effective food safety detection and control technology. Because IR covers a wide range of electromagnetic spectra (780 to 2500 nm), IR can be applied to the detection of a variety of foods including edible oil, meat, aquatic products, fruits and vegetables [138,139,140,141,142,143,144]. When IRs radiation penetrates the sample, the radiation is reflected, absorbed or transmitted by molecular bonds, resulting in the energy change of light, which can reflect some characteristic chemical bonds, thus reflecting the characteristics of the tested product [145,146]. In the application of edible oil, IR shows many abilities, such as distinguishing different kinds of oil, grading the quality of oil, detecting harmful substances in oil, etc. [138,143,147,148,149,150]. Using near infrared (NIR) technology to detect mold in edible oil has also been a research hotspot in recent years. Researchers have promoted the further application and development of IR technology by establishing qualitative and quantitative analysis models for AFB_1_ pollution in edible oil [151,152,153].

#### 4.2.3. Terahertz (THz) Spectroscopy

With the development of optical and electronic technology, terahertz spectroscopy (THz) has been a revolutionary development, and shows great potential as a new technology tool for nondestructive food testing [154,155,156,157]. As a technical information link between microwave spectroscopy and infrared spectroscopy, THz has the characteristics of both, making it widely used in basic research and industrial practice [158,159]. Like other spectral technologies, thanks to the development of chemometrics methods, THz has become a powerful technical tool in the food industry, due to its strong detection and quantification capabilities [156,157,160]. Through the combination of THz and chemometrics methods, researchers have constructed a rapid nondestructive detection model for AFB1 in edible oil. Although the accuracy is slightly lower than other conventional analysis methods, it provides a possibility for THz in food safety detection [161]. In a recent study, researchers further improved the accuracy of THz in detecting AFB1 in edible oil by adding pretreatment and other methods on the basis of predecessors, and reduced the LOD of AFB_1_ to 1 μg kg^−^^1^, and the accuracy is improved to more than 90% [161,162]. The cross integration of THz and chemometrics and other disciplines is conducive to promoting its application and development in the detection of AF_S_ in the edible oil industry. At the same time, the limitations of THz should also be clear, such as the low detection limit and sensitivity advantage are not obvious, the penetration of the detected object is limited, there is scattering effect, the technology is expensive, the database is lack, etc. [163].

#### 4.2.4. Surface-Enhanced Raman Spectroscopy (SERS)

As a complementary analysis technology of IR, the Raman spectroscopy (RS) technology is sensitive to the symmetrical vibration of covalent bonds of non-polar groups (such as C=O, C-C and S-S) [164,165,166]. Therefore, RS has the advantages of being fast, sensitive and simple in the detection and evaluation system of food [165,167,168]. However, traditional RS has some limitations, such as Raman scattering. Therefore, researchers have developed SERS signal enhancement technology represented by electromagnetic field enhancement and chemical enhancement [165]. At present, the application of SERS technology in the detection of AFs is still challenging, and the intersection of technology development and multidisciplinarity (such as materials science, stoichiometry, etc.) is the focus of researchers [165]. In recent years, researchers have reported a variety of SERS schemes for AFB_1_ detection in edible oil, such as SERS tag detection using antibodies and aptamers, sandwich immunoassay based on SERS, etc. [169,170,171,172,173,174]. The growing research results show that SERS technology is becoming a powerful tool to ensure the safety development of the food industry, especially in the safety supervision of AFs. However, it cannot be denied that challenges still exist, such as the development of targeted new materials, the optimization of key core technologies, and the practical application of research results [175,176,177].

### 4.3. Immunological Technology

#### Enzyme-Linked Immunosorbent Assay (ELISA)

In recent years, researchers have often used immunochemical methods to determine mycotoxins in food, in addition to traditional chromatographic techniques. The core of immunochemistry is the specific interaction between immunoglobulin (Igs) and antigen (Ag). Several immunochemical methods have been applied to detect mycotoxins in edible vegetable oils, such as enzyme-linked immunosorbent assay (ELISA) and biosensors based on immunoassay.

ELISA is one of the most commonly used methods for detecting mycotoxins [24]. It has been designed and developed on the basis of the principle of specific immune responses between Igs and Ags. The specificity of this immunoassay is due to the use of enzyme-labeled Igs or Ags and solid-matrix-restricted immunoglobulins to capture unlabeled silver in the analyte and detect it with labeled immunoglobulins. Although ELISA is well developed and widely used in food analysis, clinical practice, biotechnology, environmental, chemical, and other industries, it still has several deficiencies, such as excessive dependence on the matrix caused by the interaction between the target antigen and matrix components. The standard ELISA is composed of four main parts (immune-recognition element, sorbent substrate, enzyme label, and chromogenic reagent), and the deficiency of the central part is the root cause of the limitation of ELISA. In recent years, researchers have used the cross-fusion of multiple technologies to drive the performance of one of the components or the whole ELISA, especially in terms of sensitivity, accuracy, and stability [27].

For mycotoxins, due to the high singularity of ELISA, the developed kit has specific recognition ability and has been widely used in the detection of mycotoxins [70]. For example, Qi et al. [20] used ELISA and UPLC–MS/MS to detect AFB_1_ in peanut oil, although the LOD was only 1.08 μg kg^−1^, much higher than the LOD of UPLC–MS/MS (the LOD is 0.01 μg kg^−1^) [20]. It has been affirmed because of its accuracy, rapidity, and other advantages. For the actual detection of other harmful substances, such as AFB_1_, AFB_2_, AFG_1_ and AFG_2_, in different edible oils (oils of soybean, coconut, peanut, fennel, melon, and palm kernel), ELISA showed satisfactory results and the concentration was lower than the legislative limit [178,179,180]. On this basis, the researchers developed a commercial ELISA kit that can detect AFB_1_, which can be applied to a variety of samples including edible oil, and the detection limit can be as low as 3 ppb. Although the current ELISA technology or kit still has problems such being as time-consuming, high cost, and cumbersome operation, with the advancement of technology, ELISA technology shows strong application potential [27].

### 4.4. Electrochemical Biosensing Technology

Due to rapidity, small footprint, economy, sensitivity, and unique capabilities, electrochemical biosensing devices have received particular attention in assessing food quality, mainly reflecting AFB_1_ levels in food samples [181]. The AFB_1_ electrochemical biosensor can produce various types of analytical signals, such as voltage, current, and impedance [182,183]. The standard transduction methods are amperometric, electrochemical impedance spectroscopy (EIS), and voltammetry (potentiometry).

#### 4.4.1. Amperometric Biosensors

The amperometric biosensor is an electrochemical device with high selectivity and sensitivity that takes the change in the measuring current as the analysis signal. Because the change in the current is closely related to the concentration of AFB_1_ in food samples and the change can be achieved by maintaining a stable potential, an amperometric biosensor is relatively perfect. A typical amperometric biosensor consists of two- or three-electrode systems (containing a functional electrode, a reference electrode, and an auxiliary electrode), and the analytical performance of the latter is significantly higher than that of the former (Figure 5) [181]. This is because the additional auxiliary electrode not only increases the area of the detection surface but also increases the current between it and the functional electrode, as well as the operating potential between the functional electrode and the reference electrode, thereby enhancing the changes in the detection process of AFB_1_ in food in electronic dynamics. On the contrary, the dual-electrode system does not include auxiliary electrodes, which may lose their function at high temperatures. Therefore, amperometric biosensors with dual-electrode systems are not used to analyze the quality of food samples [181].

Even functional electrodes are usually made of inert metal materials (such as platinum, gold) or carbon (graphite, glassy carbon). The main drawback is reproducibility of measurements. Currently, printed electrodes have become a good substitute because their cost and mass production can be controlled [70,78].

Researchers used two kinds of nanomaterials with different charges to deposit on the electrode alternately, obtaining a multilayer electrode with a sandwich structure with excellent conductivity and rich electrochemical active sites [184]. Such a biosensor has good selectivity, reproducibility, and stability. In the subsequent optimization test, the optimized electrochemical biosensor was found to have significant stability and even after being placed for a period of time, it showed good LOD (0.002 ng mL^−1^). This sensor is believed to be one of the best biosensors for detecting mycotoxins. Researchers applied the electrochemical biosensor to detect AFB_1_ in real oil samples and found that it has a good recovery rate (98.11–103.36%).

Xuan et al. [185] developed an integrated AFB_1_ detection platform that uses disposable screen-printed electrodes (SPEs), allowing routine detection without electrode modification. According to the SPE used, the platform can simplify the tedious sample processing process through high-throughput processing, reduce operating errors, and improve experimental reproducibility, which can benefit large-scale sample processing. The detectable concentration range of AFB_1_ was 0.08–800 μg kg^−1^ with a LOD of 0.05 μg kg^−1^. Analysis of real samples and verification of the method showed the results of the new sensor to be consistent with those of the classical method (LC–MS/MS), indicating that the developed method has the potential to monitor AFB_1_ in peanut oil.

Another study reported a new aflatoxin biosensor based on the AFB_1_ inhibition of acetylcholinesterase (AChE) [186]. The core of this method is to immobilize choline oxidase on a screen-printed electrode modified with Prussian blue (PB). The electrode used in the biosensor can detect H_2_O_2_ at low potential. As per the results, the linear operating range of the biosensor is estimated to be 10–60 ppb and the LOD is 2 ppb. On using real olive oil samples to evaluate the sensor, the recovery rate was found to be 78 ± 9% at 10 ppb.

#### 4.4.2. Electrochemical Impedance Spectroscopy (EIS)

Electrochemical impedance spectroscopy (EIS) technology is an effective monitoring tool for identifying and monitoring changes in mycotoxins at the interface between electrode surface modifications. When the target analyte is combined with a biometric element, it generates an electrochemical response by changing conductivity and capacitance through an impedance biosensor [187]. These biosensors monitor the impedance changes caused by the interaction between the target detection object, such as AFB_1_, and the biometric element fixed on the working electrode, and display the detection results in the form changed electron flow on the working electrode [188,189]. Typical potentiometric sensors are also suitable for the three-electrode system.

The main parameter of the EIS is the charge transfer resistance value (RCT), which is closely related to the reaction between immobilized mycotoxins and the antibody antigen and is also proportional to the target detector/concentration of the target [65,66]. For determining AFB_1_, Yu et al. [190] reported a sensitive and convenient EIS method involving MWCNT/RTIL/Ab-modified electrodes coated in bare GCE. The experimental results show that the resistance of the MWCNT/RTIL/Ab-modified electrode (605.6 Ω) is higher than that of bare GCE (151.9 Ω). When AFB_1_ was immobilized on MWCNT/RTIL/Ab-modified electrodes, the increase in the electron transfer resistance (Ret) value was found to be directly related to the AFB_1_ amount. The specific interaction between AFB_1_ and Ab causes an increase in the Ret value, which leads to the production of electrically insulating biological conjugates, which will prevent the electron transfer process of redox probes. Therefore, the EIS measurement results are consistent with the above cyclic voltammetry results. Because of its simple characteristics, this method can be widely used to detect various agricultural products and edible oils.

For many researchers exploring mycotoxin detection methods, aptamer-based EIS has become a hot research topic. Aptamer-based impedance biosensors have achieved satisfactory results in detecting mycotoxins in food and have great potential for practical application in edible oils.

#### 4.4.3. Voltammetry Biosensors

Voltammetric biosensors solve the problem of obtaining analytical data using ion-selective electrodes. Similar to amperometric biosensors, voltammetry also requires a two- or three-electrode system. When the current is constant, it can detect target analytes, such as AFB_1_, in food samples by evaluating the change in circuit potential between the functional electrode and the reference electrode [60,191].

Biosensors have also shown promising results in detecting the AFB_1_ content in edible oils. For example, Wang et al. [192] developed a new disposable electrochemical biosensor based on stripping voltammetry to detect copper ions released from copper apatite. The biosensor uses copper ions as a signal label to immobilize AFB_1_ antibody on a screen-printed carbon electrode (SPCE) modified by gold nanoparticles. The detection is performed by the voltammetric signal of the dissolution of copper ions released from acid hydrolysis of copper apatite, and copper apatite increases the number of loaded copper ions. The electrochemical signal is further amplified. Peanut oil was used to evaluate the reliability and application potential of biosensors. Researchers believe that this new method will be applied to many fields in the near future because of its many excellent characteristics (low cost, rapidity, accuracy, and high sensitivity).

#### 4.4.4. Nanomaterial-Based Biosensors

Recently, different nanomaterials, such as carbon and metal, have been used to modify the active surfaces of macroelectrodes and microelectrodes to design electrochemical biosensors for the detection of AFB_1_ [193,194,195]. This is because new biosensors directly use nanomaterials or other materials containing nanoparticles that show significant characteristics, such as high sensitivity and specificity for detecting targets, reliability, and consistency of products [181,196,197]. Nanomaterials significantly increase the effective surface area of biosensors and further improve the analytical performance [60,194]. Nanomaterials also enhance some characteristics of biometric elements in biosensor devices in terms of electrical, catalytic, optical, and thermal properties [198]. According to previous studies, some of the key functional enhancements are the enhanced immobilization of biomolecules, generation and expansion of analytical signals, and enhanced usability of fluorescent labels.

##### Characteristic of Nanomaterials Based Electrochemical Biosensors

The role of nanomaterials in biosensors is mainly reflected in the immobilization of biomolecules, signal generator, fluorescent labeling, and signal amplification.

Nanomaterials not only immobilize biomolecules but also increase the interaction between different molecular materials. In addition, nanomaterials enhance the stability of biomolecular immobilization, thereby increasing the signal strength of the immunoassay [31]. Metal nanomaterial particles, such as AgNPs and MOFs, can increase the surface area and biocompatibility of biomolecules bound to the detection target. However, non-metallic nanomaterials show negatively charged functional groups, which can be used as an effective carrier to bind and fix with positively charged targets.

##### Signal Generator

Xue et al. [31] reported that, when the photoelectric signal changes, nanoparticles such as gold and silver can act as a signal generator. By adjusting the fluorescence signal generated by nanomaterials, a new AFB_1_ nanoprobe can be constructed. In addition, because these nanoparticles can be prepared in different sizes according to need, they have good functionality, stability, and scalability [133,199].

##### Fluorescent Label

Nanomaterials have unique optical properties that enable them to be widely used in a variety of disciplines, especially in the detection of hazardous substances in food. Nanomaterials can detect AFB_1_ by sensing optical signals (absorbance, chemiluminescence, fluorescence, etc.) [31]. Some nanomaterials, such as metal nano-ions and quantum dots, have been used as fluorescence quenching agents because of the ability of AFB_1_ to directly quench or reduce the fluorescence intensity. In addition, quantum dots have transformed fluorescein into a marker element that binds to aptamers or antibodies.

##### Signal Amplification

Nanomaterials can also be used as functional materials for various electrodes, signal components, etc., to amplify signals in various ways. For example, on the electrode surface of electrochemical sensors, nanomaterials such as gold and silver can amplify the analytical signal by enhancing the redox reaction. Some metal nanoparticles, such as gold, can amplify signals related to their characteristics, such as unique catalytic activity, biocompatibility, and multiple absorption sites. Carbon, graphene, and other non-metallic nanomaterials improve the analytical performance by increasing the surface area.

### 4.5. Bioinspired Recognition Elements for Biosensors

A biosensor is independent quantitative analysis equipment used to study the analytes required in different types of food samples. A biosensor consists of many parts [29,30] (Figure 6). Biometric elements are the core components of biosensors and can detect specific target analytes. The quality of biometric elements usually determines the specificity and sensitivity of analysis [200,201]. Biorecognition elements, including antibodies, aptamers, molecularly imprinted polymers, and enzymes, have been used to manufacture biosensors [34,64,202]. These elements show increased sensitivity and selectivity for target analytes. Critical biometric elements for developing biosensors to detect AFB_1_ in edible vegetable oil are elaborated below.

#### 4.5.1. Antibody

Antibodies have been used as recognition elements for developing biosensors because of their specificity and sensitivity [200]. Biosensors that use antibodies as recognition elements are called immunosensors, and their mechanism relies on the specific recognition of aflatoxin epitopes by antibodies.

The first batch of polyclonal antibodies, developed in 1976, became the basis for most mycotoxin detection methods. In the following decades, polyclonal and monoclonal antibodies were the basis for most mycotoxin detection methods [192,200,203]. Today, in addition to monoclonal and polyclonal antibodies, various other types of antibodies are being used to detect target analytes. Researchers have developed an antibody-based immunosensor that can directly recognize AFB_1_ and is used in peanut oil with a concentration range of 0.001 to 100 ng mL^−1^, with a detection limit of 0.2 pg mL^−1^ [192].

However, the production of monoclonal antibodies and polyclonal antibodies is complex and the antibodies degrade, denature, and aggregate easily [204,205]. In recent years, with the development of protein- and DNA-based new engineering technology, it has become possible to develop modified and recombinant antibodies (RAbs). RAbs integrate many advantages of biosensors, such as simple operation and a high degree of automation, high throughput screening, low requirements for configuration attributes, and the trend of more miniaturization [200]. Zhao et al. [206] developed a novel method of MB-dcELISA for AFB_1_ based on the mimotope of an RAb and nanobody. This study effectively proved that compared with monoclonal antibodies, an RAb is more economical and easier to prepare. Compared with chemically synthesized toxic antigens, immunoassay is safer and performs better in validation studies. In real samples (corn germ oil and peanut oil), the LOD of AFB_1_ is as low as 0.13 ng mL^−1^. Other researchers also designed an RAb with increased sensitivity to low-molecular-weight haptens, and this RAb was validated in olive oil with a lower LOD (0.03 ng mL^−1^) for AFB_1_ [190].

Researchers recently found that, by increasing the immobilization of antibodies and giving full play to the characteristics of specific antibodies, the performance of the sensor can be effectively improved, and on this basis, some immunosensors have been developed for detecting AFB_1_ in edible oil [133,196]. For example, to determine AFB_1_, Shi et al. [207] proposed a novel immobilized immunosensor based on graphene supported with hybrid gold nanoparti-cles-poly4-aminobenzoic acid. In the study, after the reduction in graphene oxide by PABA via an epoxy ring opening reaction, the nanocomposite PABA-r-GO was obtained. Then, gold nanoparticles (AuNPs) were prepared on this basis to form a Au-PPABA-r-GO nanohybrid. The final sensor was obtained by the covalent binding of the COOH group of functional nanocomposites with an AFB_1_-specific antibody. The sensor has good performance (linear range 0.01–25 ng mL^−1^ and LOD 0.001 ng mL^−1^) and has been successfully applied to detect real vegetable oil. This sensor also has good reproducibility and selectivity, especially stability, and can be stored at a low temperature for a long time.

#### 4.5.2. Aptamers

Aptamers are single-stranded RNA or DNA (20–90 oligonucleotide sequences with specific sequences) that can bind to various targets, such as ions, antibodies, proteins, cells, and organic molecules [208]. The particular recognition ability of aptamers relies on the three-dimensional structure of a high-affinity target-induced DNA three-dimensional structure. The researcher procured specific targets for aptamers by screening oligonucleotides using the Phylogenetic Evolution of Ligands for Exponential Enrichment (SELEX) program. Aptamer sensors are biosensors integrated with aptamers developed in the 1990s [200,208,209,210,211].

Recently, aptamers have attracted significant attention in food contamination analysis and are used for various sensing applications due to their inherent benefits: (1) aptamers are obtained from in vitro synthesis, so animals are not necessary; (2) aptamers have lower toxicity, immuno-genicity, and production cost; (3) aptamers have enhanced chemical and thermal stability; (4) aptamers have excellent batch-to-batch reproducibility; (5) aptamers have a smaller size and show a remarkable ability to penetrate the tissue and adhere to target molecules; and (6) it is possible to change their structure [193,212,213].

Notably, the immobilization of aptamers is a critical step in biosensor design because it can affect the affinity of aptamers for their targets and their long-term stability in fundamental sample analysis.

Therefore, researchers have developed many strategies to immobilize aptamers: (1) adsorption or π–π stacking interactions between DNA bases and modified graphene oxide (GO) interface [214], (2) aptamers with carboxylic acids on surfaces or nanomaterial covalent bonding of groups [184,215], (3) binding of sulfide aptamer with CdTe quantum dots (QDs) or gold-based materials [216], (4) binding with avidin or other affinity interactions based on biotin streptomycin affinity [200,217,218], and (5) hybridization with partially complementary single-stranded DNA previously fixed on the surface of nanoparticles [219,220,221,222].

Table 2 describes some examples of aptasensors recently reported for the detection of AFB_1_ in edible oils. About half of the previous reports have been based on fluorescent mycotoxin aptamer sensors. Some of them use metal or nanostructured materials, such as gold nanoparticles (AuNPs), GO, single-walled carbon nanotubes, or TiO_2_ tubes, and are used to prepare aptamer sensors.

Nanometer material has always been the focus of research, and its applications in biosensors are also diverse. Black phosphorus nanosheets (BPNSs) have great application prospects in biosensors due to their unique characteristics [223]. Wu et al. [224] developed a highly specific and sensitive aptamer sensor (UCNPs-BPNSs) based on the team’s research on upconversion nanoparticles (UCNPs) [196]. The research team attached UCNPs to the surface of BPNSs at a very small space distance (less than 10 nm) through glutaraldehyde crosslinking method and π-π stacking effect method, and then constructed the fluorescence resonance energy transfer (FRET) system. This aptamer sensor can effectively detect AFB_1_ in peanut oil and other foods quantitatively with good linear range (0.2–500 ng mL^−1^) and LOD (0.028 ng mL^−1^).

Xia et al. [225] proposed a label-free, single-tube, homogeneous, and inexpensive assay for AFB_1_ based on fine-tunable double-ended stem aptamer beacons (DS) and the effect of aggregation-induced emission (AIE). The structure of the DS aptamer beacon can provide end protection against exonuclease I (EXO I) to the aptamer probe and endow it with specificity and a rapid response to the target AFB_1_. Compared with the traditional molecular beacon structure, the stability of the DS aptamer beacon can be adjusted by adjusting its two terminal stems so that the affinity and selectivity of the probe can be precisely optimized. Using an AIE-active fluorophore, which is illuminated by the aggregation of negatively charged DNA, AFB_1_ can be measured label-free. The method has been successfully applied to the analysis of AFB_1_ in peanut oil, with a total recovery of 93.59–109.30%. Therefore, beacon-based DS assays may help in real-time monitoring and control of AFB_1_ contamination.

Yang et al. [226] first devised a selection method based on rational truncation and post-splicing and developed a bivalent anti-AFB_1_ chimeric aptamer (B72) that was measured by micro-thermophoresis (MST) compared to the initial selection. The affinity of the anti-AFB_1_ aptamer (B50) increased by 188-fold, and the study also found that B72 has a dual binding site for AFB_1_, which is consistent with the experimental results obtained by isothermal titration calorimetry (ITC) and molecular docking simulations. Therefore, on the basis of the peroxidase-like activity of gold nanoparticles catalyzing 3,3,5,5-tetramethylbenzidine (TMB), an aptamer sensor of gold nanoparticles (AuNPs) was developed by the colorimetric detection of AFB_1_. The assay further validates the practical applicability of the chimeric aptamers. The aptasensor could identify AFB_1_ with an excellent linear range (5–5120 nM) and detection limit (1.88 nM) in the corn oil environmental test of H_2_O_2_. Therefore, this study can be called a general selection method for designing high-affinity aptamers and constructing novel aptamer-based biosensing platforms for high-sensitivity and specificity analysis of other targets.

Zhong et al. [227] manufactured an electrochemical aptamer sensor in a similar way for the sensitive detection of AFB_1_. The researchers used electrodeposited AuNPs to prepare AuNPs/ZIF-8 nanocomposites on glassy carbon electrodes (GCEs) decorated with the eight zeolite imidazolate framework (ZIF-8), which increased the surface area of the electro desorption molecular load. Compared with other previously reported sensors, the aptasensor developed under optimized conditions shows a more comprehensive linear range (10.0–1.0 × 10^5^ pg mL^−1^) and a lower detection limit (1.82 pg mL^−1^). In addition, the constructed aptasensor possesses excellent selectivity, reproducibility, and stability. Moreover, the aptamer sensor has been successfully used to detect AFB_1_ in corn oil and peanut oil samples, and the recovery was between 93.49% and 106.9%, which proves the potential application value of this method. Researchers are very interested in this kind of electrochemical aptamer sensor. Wang et al. [228] also developed an AFB_1_ electrochemical aptamer sensor for detecting peanut oil in a similar way. The difference lies in the use of different composite materials (zinc and nickel bimetallic organic skeleton materials).

The hybridization chain reaction (HCR) is a commonly used isothermal nucleic acid amplification technique, and due to the characteristics such as no enzyme, high amplification efficiency etc., HCR is usually used as a new synthetic material technology and is widely used in various sensors. Researchers have fully combined the characteristics of HCR to build a signal amplification strategy, which has been successfully applied to the sensitive detection of AFB_1_ [229,230,231,232,233,234,235]. Wang et al. [236] proposed a fluorescent aptamer sensor based on DNA walker, DNA tetrahedral nanostructures (DTNs) and network HCR. Among them, DNA walker was used as the signal amplifier induced by AFB_1_ target, and combined with self-assembled DTNs. Finally, based on network HCR, signal amplification is realized and sensitive detection of AFB_1_ in peanut oil was realized with with LOD of 0.492 pg mL^−1^ and the linear range of 1–1000 pg mL^−1^. In the other report, Zuo et al. [237] combined DNAzyme with substrate chain (Zn-Sub) and enzyme chain (Zn-Enz) with HCR products to form a Y-shaped structure, which can significantly enhance the fluorescence intensity of the detection target. The fluorescent aptamer sensor proposed by researchers shows excellent performance with LOD of 0.22 nmol L^−1^ and the linear range of 0.4–16 nmol L^−1^.

The emerging quantum dots (QDs), represented by carbon quantum dots (CQD_S_), graphene quantum dots (GOD_S_) etc., have attracted great attention and are widely used in various sensor fields since their discovery because of their excellent optical properties, low toxicity, stability and low cost etc. [238,239,240]. QDs-based sensors can adopt different working mechanisms and be applied to detect different substances, including AF_S_ in edible oil [51,173,238]. According to the characteristics of QDs, Xuan et al. [185] and Ye et al. [241] developed and constructed different magnetic control pretreatment platforms, which were actually applied to the detection of AFB_1_ in peanut oil and agricultural products, and both showed good detection characteristics. Other researchers used a quencher system composed of quantum dots and graphene oxide to detect AFB_1_ in peanut oil, which also showed good detection characteristics [242].

##### SERS

As mentioned above, SERS is a promising analytical tool with many advantages over traditional AFB_1_ detection methods, including high sensitivity, easy sample preprocessing, and non-destructive testing [166,174,243]. Compared with antibodies, aptamers have the advantages of low cost, easy synthesis, good stability and strong specificity to target molecules. With the mature development of aptamer manufacturing technology, they have gradually become one of the most potential recognition elements in SERS labeling detection.

Recently, several authors have combined advanced composite materials with SERS aptamer sensors to develop new procedures for AFB_1_ detection. For example, on the basis of the combination of a multifunctional capture probe (Fe_3_O_4_@Au report the strong Raman signal of probe 1 (AU)-4MBA@AgNSs-Apt), an ultrasensitive assay was successfully developed for a high-performance SERS aptamer sensor of AFB_1_. He et al. [174] reported that, in the presence of AFB_1_, the probe was released from the capture probe, resulting in a decrease in SERS intensity, possibly due to the specific binding affinity between the aptamer and AFB_1_. For AFB_1_ detection, a wide linear range, from 0.0001 to 100 ng mL^−1^, was obtained, with an R^2^ of 0.9911, and the LOD was calculated as 0.40 pg mL^−1^. Finally, after extracting AFB_1_ from peanut oil samples, the SERS aptamer sensor was successfully applied to the analysis of AFB_1_, and the recovery was between 96.6% and 115%. Therefore, the novel SERS aptamer sensor is a promising analytical tool for detecting AFB_1_ in actual samples.

In the report by Yang et al. [169], with the help of the specific interaction between AFB_1_ and aptamer, a novel SERS-based universal aptamer sensor platform was constructed to detect AFB_1_. First, gold nanotriangle (GNT)-DTNB@Ag-DTNB nanotriangles (GDADNTs) were synthesized and used as SERS active substrates. These magnetic beads and amino-terminal-aptamer-conjugated magnetic beads (CS-Fe_3_O_4_) were then used as capturer and reporter of AFB_1_, respectively. Finally, the platform showed excellent sensitivity under optimized assay conditions, with a lower LOD (0.54 pg mL^−1^) and a more comprehensive linear range (0.001–10 ng mL^−1^). In addition, the high stability of SERS substrate activity was maintained for at least three months, with an RSD of ~5%, which has good selectivity for general coexistence interference. The excellent sensitivity and selectivity of micro-AFB_1_ detection are mainly due to the substantial Raman-enhancing effect of GNTs as the core of GDADNTs, which results from the bilayer of reporter molecules, aptamer specificity, and the super-paramagnetic CS-Fe_3_O_4_, respectively. The researchers also evaluated and confirmed that the established SERS aptamer sensor can be used to detect AFB_1_ in peanut oil samples.

In a subsequent study, on the basis of previous research, another simple and sensitive SERS aptamer sensor was developed for detecting AFB_1_ in peanut oil [170]. In this study, the researchers used an aminoterminal AFB_1_ aptamer (NH2-DNA1) as a SERS aptamer sensor, magnetic beads conjugated to a thiol-terminal-complementary AFB_1_ aptamer (SH-DNA2) (CS-Fe_3_O_4_) as enrichment nanoparticle probes, and AuNR@DNTB@Ag nanorods (ADANR) as reporter nanoprobes. 5,5′-Dithiobis (2-nitrobenzoic acid) (DNTB) is embedded in gold and silver core/shell nanorods as a Raman reporter molecule, which has a large Raman scattering cross section and no fluorescence interference. Furthermore, CS-Fe_3_O_4_ has good biocompatibility and superparamagnetism, which can quickly enrich signals. Therefore, NH2-DNA1-CS-Fe_3_O_4_ and SH-DNA2-ADANRs were prepared by a mixed reaction between aptamers and complementary aptamers. When present, AFB_1_ will compete with NH2-DNA1-CS-Fe_3_O_4_ to induce SH-DNA2-ADANRs to dissociate from CS-Fe_3_O_4_, further reducing SERS signals. According to the SERS aptamer sensor, the lower detection limit of AFB_1_ is 0.0036 ng mL^−1^ and the correlation coefficient is as high as 0.986. The effective linear detection range is 0.01–100 ng mL^−1^, obtained with a correlation coefficient as high as 0.986. Finally, the specificity and accuracy of the SERS aptasensor were proved by detecting AFB_1_ in natural peanut oil.

Similar research strategies are reflected in other reports. Jiao et al. [244] developed a gold-silver core-shell nanoparticles (Au@Ag CSNPs) SERS sensor decorated with 5-aminotetramethylrhodamine (NH_2_-Rh). Based on the optimization of experimental conditions, the sensor can be combined with solid phase extracts of peanut oil, hazelnut, and other samples to achieve a quantitative analysis of AFB_1_ with detection range and LOD 0.1–5.0 ng·mL^−1^ and 0.03 ng·mL^−1^, respectively.

Various AFB_1_ sensors are also identified in edible vegetable oil by electrochemical detection, which has some unique advantages, such as low cost, high sensitivity, and the possibility of micromachining. For example, Xiong et al. [245] revealed a highly innovative method based on dual-DNA-tweezer nanomachines to detect AFB_1_ in olive and peanut oils. Wu et al. [246] presented a method based on ferrocene and β-cyclodextrin (β-simple electrochemical aptamer sensor for host–guest recognition between CD) to detect AFB_1_ in peanut oil, with a low LOD (0.049 pg mL^−1^).

#### 4.5.3. Molecularly Imprinted Polymers (MIPs)

MIPs have been used as recognition elements to develop biosensors, and synthetic polymers have displayed precise target recognition [133,202,247]. These artificial materials can recognize specific targets in complex mixtures because of specific recognition sites for binding or catalysis and functional groups with shapes and geometries complementary to those of the template molecule. These polymers self-assemble with template molecules and active/functional monomers through the polymerization of cross-linking agents. Therefore, when the template molecule is removed, pores with multiple active sites appear in the polymer, which match the spatial configuration of the template molecule [248,249]. In recent years, traditional MIPs have been applied in many cross fields, such as chromatography, drug delivery, solid-phase extraction, controlled release, bioremediation, and sensors [200,250,251,252,253,254,255,256]. In AFB_1_ detection studies, MIP-based biosensors have shown many advantages, such as unique selectivity, sensitivity, user-friendliness, and cost-effectiveness [32,42,200,202]. For instance, Li et al. [173] exploited MIPs by preparing an electrochemiluminescence (ECL) platform for AFB_1_ detection with an ultra-low LOD, of 8.5 fg mL^−1^, and a wide linear range (10^−5^ to 10 ng mL^−1^). While the MIP–ECL platform was used, the recovery rate of corn oil samples was close to that obtained by HPLC, indicating the reliability of the sensor and its potential in food safety evaluation. It is worth mentioning that, as of the publication of this review, this is the lowest LOD of AFB_1_ in edible oil.

However, MIP-based biosensors also have some disadvantages, such as generally poorer affinity and specificity than antibodies, slower binding kinetics than biological receptors, incomplete template elimination, and lower utilization of binding sites [133,200,202]. Therefore, there is increasing interest in developing improved MIPs [257,258,259]. The key to the success of the sensor of an MIP is whether the MIP is effectively attached to the transducer. Three commonly used immobilization methods are in situ polymerization, electropolymerization, and physical coating. Additionally, the number of applications of MIP sensors for detecting AFB_1_ in edible vegetable oil is limited [200,202,260].

## 5. Conclusions and Perspectives

Mycotoxin contamination, especially AFB_1_ contamination in edible oil, is usually unavoidable. A more sensitive and rapid sensor-based early warning tool for AFB_1_ detection would help to reduce risk. Various traditional, modern, and biosensing technologies have been used to detect toxins in contaminated food. Spectroscopic techniques, chromatographic techniques are general methods for the detection of AFB1 in edible oils. In recent years, based on the cross-integration of multiple disciplines, the innovation, progress and development of general methods have also been promoted. Although traditional chromatographic techniques can effectively detect mycotoxins, their performance in all aspects cannot achieve satisfactory results. Combined use with other sensor equipment can effectively improve reliability, sensitivity and accuracy. However, due to the high cost of equipment, on-site inspection cannot be performed, and sample pretreatment is required, which limits the use of chromatography technology in the detection of AFB1 in edible oil. The development of spectroscopic techniques has become increasingly diverse and can effectively detect mycotoxins, especially AFB1 in edible oils. However, these methods are not suitable for on-site detection, because they still have many shortcomings, such as low sensitivity and reliability, and the need for professional personnel to operate.

Unlike conventional detection techniques, novel biosensors show high accuracy, sensitivity, and specificity; better cost controllability and portability; and reliability and simplicity in operation.

This review also discusses the development of important recognition elements in sensors. The recognition element of the sensor should have sensitivity and specificity sufficient enough to detect small amounts of target toxins, even in samples with complex matrix systems. The development and use of nanomaterials further improve the efficiency of biosensor conversion systems, but these require further improvements in their sensitivity, selectivity, and reproducibility. Of course, the stability and cost will also affect the selection of identification elements, which can improve the practicability.

Despite significant progress in biosensors for the detection of AFB_1_, there are some problems and challenges in the future. (1) The recognition elements of biosensors (such as metal nanoparticles, quantum dots, and graphene) improve the efficiency of sensing systems, but all these require further improvements in terms of sensitivity, selectivity, and reproducibility. (2) Future studies can perform AFB_1_ toxicity measurements and develop advanced nanomaterial-integrated biosensors to improve the overall detection of harmful substances, such as AFB_1_, in contaminated food samples. (3) When detecting AFB_1_ in contaminated food samples, researchers can focus on combining biosensing systems with microarray technology to fabricate more portable devices. (4) Reagent-free, clean-free, calibration-free, or nonbiological contamination biosensors for aflatoxin analysis require more effort and will reduce the possible future hazards.

## Figures and Tables

**Figure 1 molecules-27-06141-f001:**
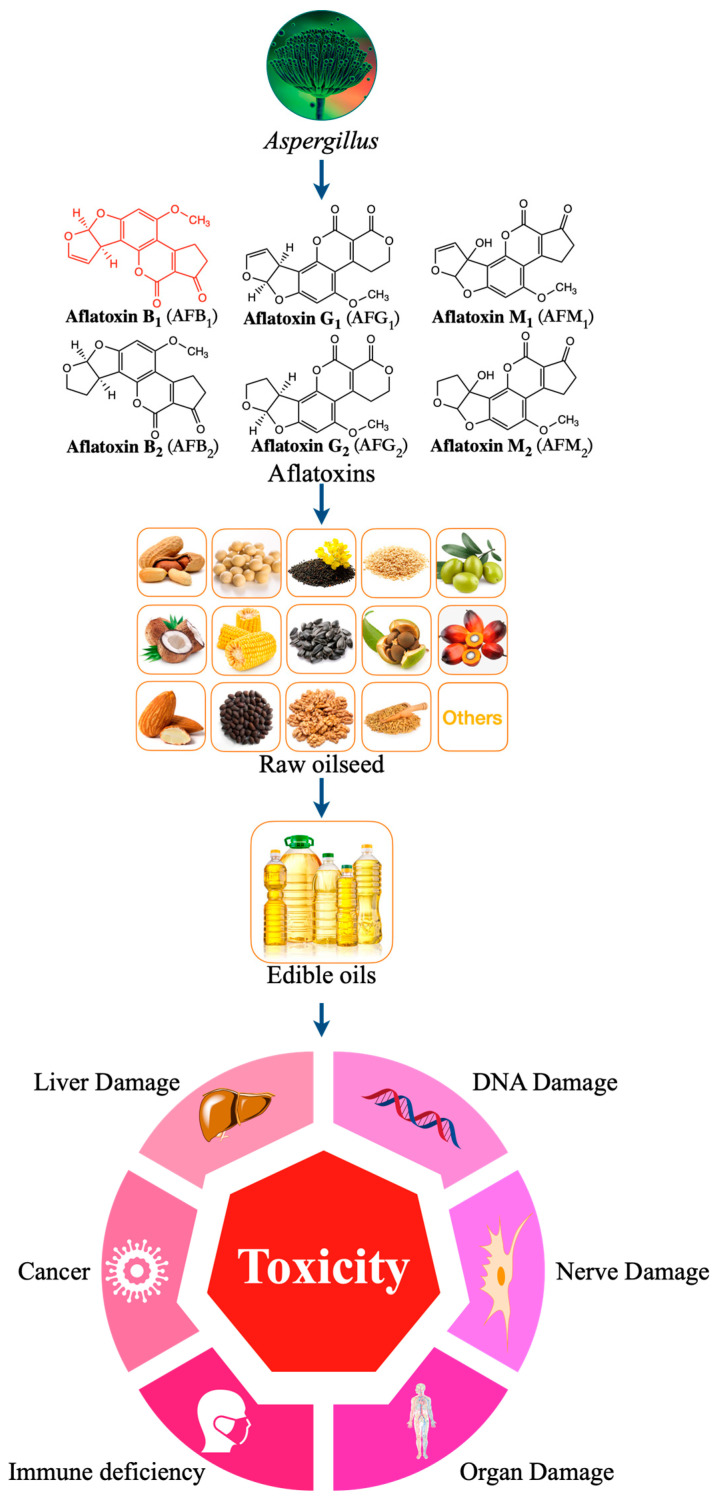
Harmful effects of different types aflatoxins contaminated edible oil.

**Figure 2 molecules-27-06141-f002:**
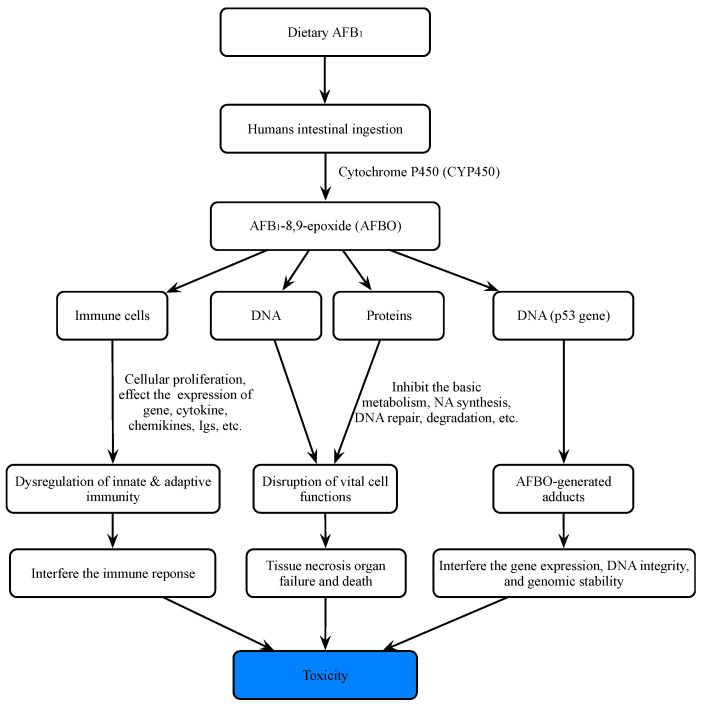
Main mechanisms of toxicity of aflatoxin B_1_ for humans.

**Figure 3 molecules-27-06141-f003:**
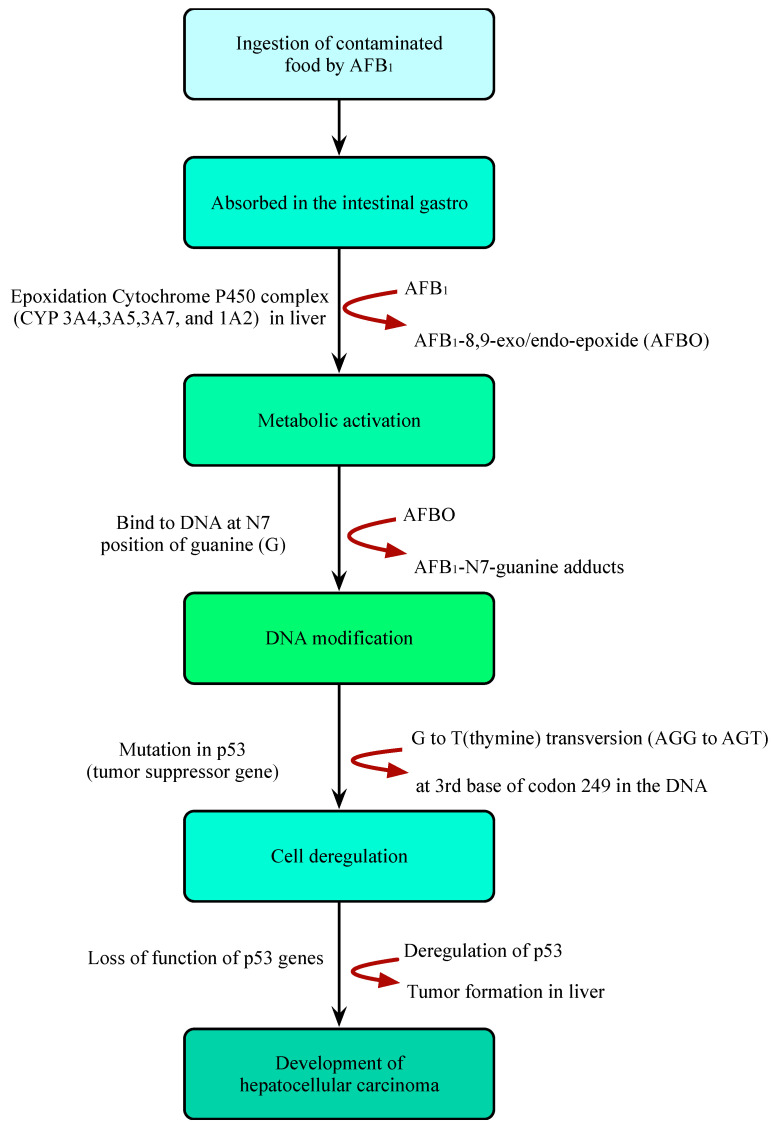
Illustration of the mechanism of hepatocellular carcinoma caused by ingestion of AFB_1_-contaminated foods.

**Figure 4 molecules-27-06141-f004:**
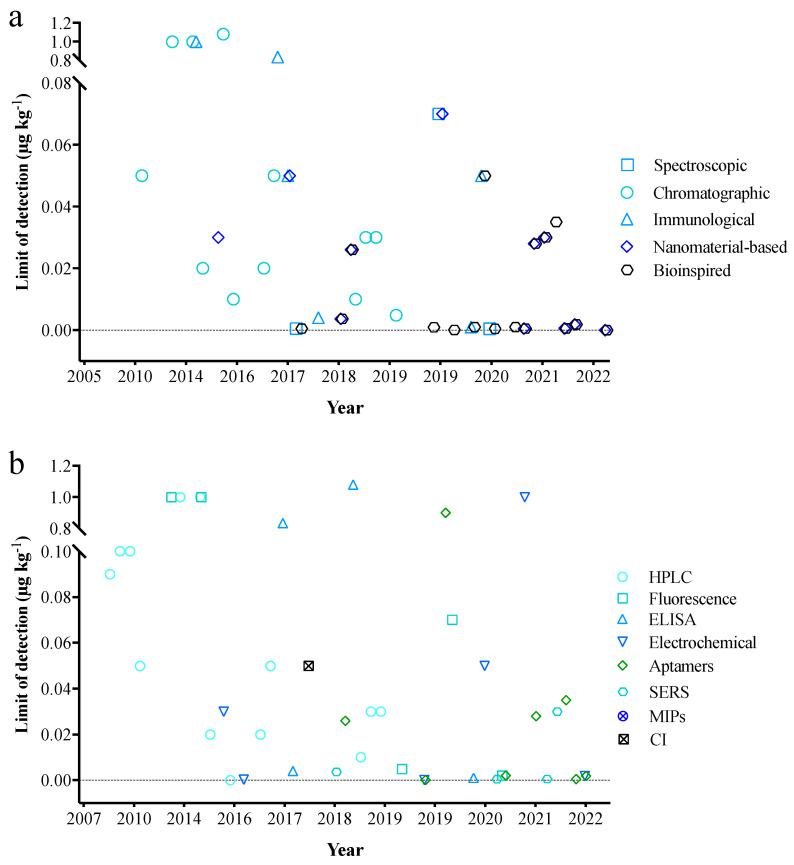
Timeline of the limit of detection on AFB_1_ in edible oil by different (**a**) detection technology and (**b**) detection method. CI: Chemiluminescence immunoassay.

**Figure 5 molecules-27-06141-f005:**
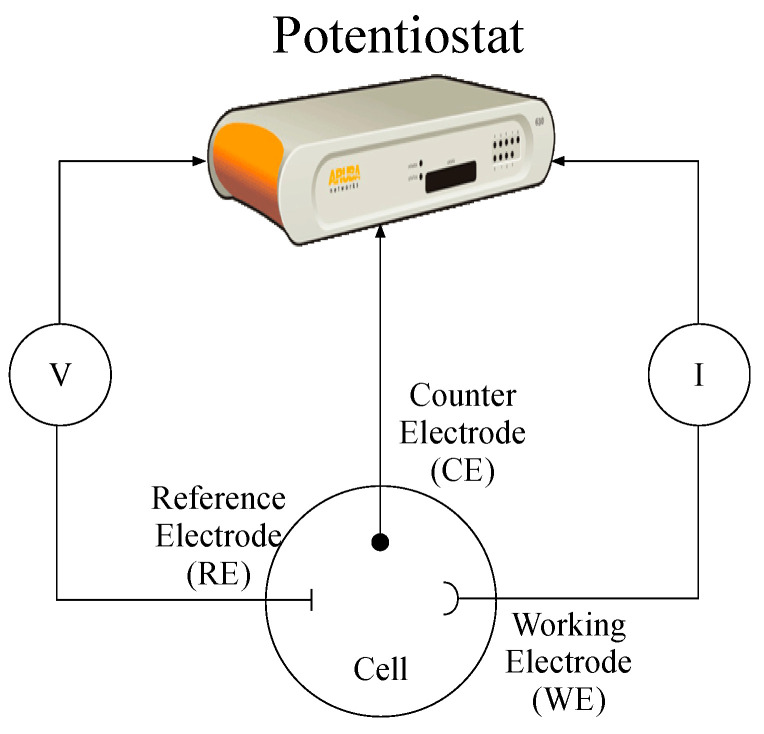
Scheme of the two or three-electrode setup used in electrochemical methods.

**Figure 6 molecules-27-06141-f006:**
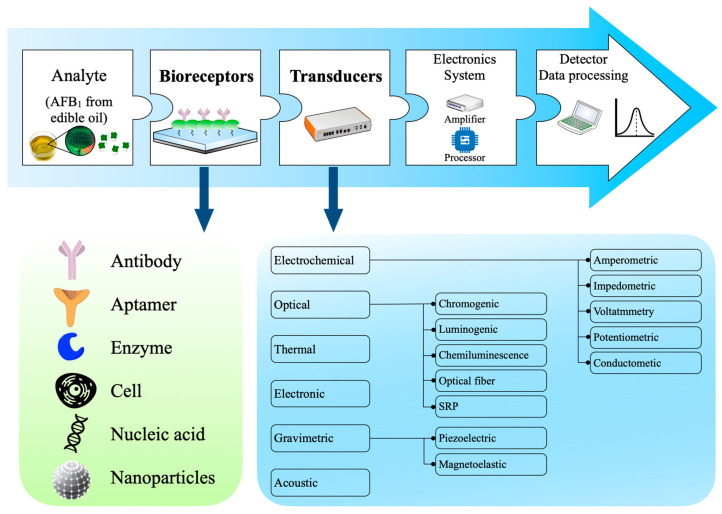
Schematic diagram of typical biosensor. Including analyzer, bioreceptor, transducer, electronic system (amplifier and processor), detector (for data processing).

**Table 1 molecules-27-06141-t001:** The maximum limits (μg kg^−1^) established for major AFB_1_ in some countries/regions for edible oils.

Countries/Agencies	Food Products	Edible Vegetable Oil	Total of AFs (μg kg^−1^)	AFB_1_ (μg kg^−1^)	Refs.
EU	Oil seeds	-	15	8	[32,69]
EU	-	Peanut oil Others oil	4	2	[32,69]
China	-	Maize oil Peanut oil Others oils	- - -	20 20 10	[73]
Greece	-	Olive oil	-	-	[69]
Russia	-	Vegetable oil	-	-	[64]
France	-	Vegetable oil	-	5	[64]
Kenya	-	Vegetable oil	20	-	[64,71]
Taiwan	-	Edible oil	10	-	[65]
Morocco	-	Vegetable oil	-	5	[72]
Thailand	All foods	Oil and fats	20	-	[64,65,74]
USA	All foods	-	20	-	[64,71,75]
Brazil	All foods	-	-	15	[76]
India	All foods	-	-	30	[65]
Chile	All foods	-	-	-	[76]
Indonesia	All foods	-	35	20	[65]
Singapore Australia	All foods	-	5	-	[64,65]
Malaysia	All foods	-	35	-	[65]
Japan Vietnam	All foods	-	10	-	[65]
Sri Lanka	All foods	-	30	-	[65]

**Table 2 molecules-27-06141-t002:** Techniques used for the detection of AFB_1_ in different types of edible oil.

Matrix	Analytical Method	Sample Preparation Method	Linear Range	Recovery	LOD	Ref.
Peanut oil	ELISA	Immunoaffinity column cleanup	-	84.40–92.60%	1.08 μg kg^−1^	[20]
Coconut oil	HPLC-FLD	Immunoaffinity chromatography	-	-	0.01–0.04 μg kg^−1^	[81]
Peanut oil Corn oil Soybean oil	HPLC-FLD	DLLME with in situ derivatization	0.1–100 ng mL^−1^	106.90–121.50%	0.03 ng mL^−1^	[82]
Sunflower oil Olive oil Canola oil Frying oil Blend oil	HPLC-FLD	Immunoaffinity column cleanup	0.04–0.16 ng g^−1^	95.56–102.13%	0.16 ng g^−1^	[83]
Vegetable oil	HPLC-FLD	Immunoaffinity chromatography - Solid phase extraction	-	95.20–99.00%	0.25 μg kg^−1^	[84]
Coconut oil Almond oil Soybean oil Olive oil	HPLC -FLD	Immuno Affinity chromatography combined with DLLME	0.005–10.00 ng mL^−1^	96.00–109.90%	830 ng mL^−1^	[85]
Sesame oil Sunflower oil Peanut oil Mixed oil	HPLC-FLD	Liquid-Liquid Extraction	0.34–109.2 µg kg^−1^	83.00–96.00%	0.09–1.5 µg kg^−1^	[86]
Canola oil Corn oil Olive oil Peanut oil Soybean oil	LC-MS/MS	Liquid-Partitioning	-	101.00–111.00%	0.030 μg kg^−1^	[87]
Soybean oil Corn oil Rice bran oil	LC-MS/MS	QuEChERS × DLLME	-	70.70–76.00%	–	[88]
Blend oil Peanut oil Maize oil	LC-MS/MS	Hollow fiber liquid phase microextraction	0.1–500 μg kg^−1^	78.59–80.61%	0.02 μg kg^−1^	[89]
Sunflower oil Palm oil Corn oil	LC-ESI-MS/MS	QuEChERS	0.04–2000 ng g^−1^	87.90–106.60%	0.01 ng kg^−1^	[90]
Soybean oil Corn oil Peanut oil Blended oil	LC–MS/MS	Immunoaffinity chromatography	0.16 μg kg^−1^	87.40–97.30%	0.05 μg kg^−1^	[96]
Olive oil Peanut oil Sesame oil	LC-MS/MS	Immunoaffinity column cleanup	2–20 mg kg^−1^	87.70–102.20%	0.1 μg kg^−1^	[97]
Olive oil	LC/ESI-MS/MS	Matrix Solid Phase Dispersion	0.2–0.4 (pg inj)	95.00–98.00%	0.2 pg inj	[261]
Olive oil Sunflower oil Soybean oil Corn oil	UHPLC-QqQ-MS/MS	QuEChERS	0.5–25 μg kg^−1^	96.00–107.90%	0.5 μg kg^−1^	[100]
Sesame oil Groundnut oil Cottonseed oil	HPLC	Liquid-Liquid extraction	0.2–0.8 μg kg^−1^	-	0.1 μg kg^−1^	[102]
Vegetable oils	GPC-HPLC-FLD	Liquid-Liquid Extraction	1.0–30.0 μg kg^−1^	82.60–90.60%	1.0 μg kg^−1^	[104]
Peanut oil Sunflower oil Olive oil	IAC-LC-ESI–MS/MS	Liquid-Liquid Extraction	0.02–10 μg kg^−1^	84.00–99.00%	0.02 μg kg^−1^	[105]
Virgin olive oil	HPLC-FLD	Solid Phase Extraction		65.50–87.50%	0.25 ng g^−1^	[110]
Canola oil Soybean oil Corn oil Olive oil Peanut oil	FL	LTC-IMSPE	0.0048–0.0126 ng·g^−1^	79.60–117.90%	0.0048 ng·g^−1^	[115]
Rapeseed oil Peanut oil Blended oil Blended olive oil Sunflower oil Tea oil Rice oil Corn oil Sesame oil Soybean oil	HPLC-MS/MS	QuEChERS	0.2–20 ng mL^−1^	87.80–98.60%	0.05 ng g^−1^	[116]
Soya bean oil Groundnut oil Beniseed oil Palm kernel oil Melon oil Coconut oil	ELISA	Immunoaffinity column cleanup (226 Aflatoxin clean-up Column)	-	-	≤0.8352 μg L^−1^	[179]
Peanut oil Virgin olive oil	ELISA& TSA-ELISA	Liquid-Liquid Extraction	-	81.40–118.80%	0.004 ng mL^−1^	[180]
Olive oil	Amperometric biosensor coupled with AChE enzyme	Liquid-Liquid Extraction	10–60 ppb	76.00–78.00%	2 ppb	[186]
Olive oil	EIS based on MWCNTs/RTIL composite films	Liquid-partitioning	0.1–10 ng mL^−1^	96.00–116.00%	0.03 ng mL^−1^	[190]
Peanut oil	Disposable electrochemical immunosensor with Au NPs modified SPCE	-	0.001–100 ng mL^−1^	90.00–102.00%	0.2 pg mL^−1^	[192]
Peanut oil	Fluorescence spectroscopy based on UCNPs upconversion nanoparticles (UCNPs)	Liquid-partitioning	0.2–100 ng mL^−1^	92.80–113.40%	0.2 ng mL^−1^	[196]
Oil	Chemiluminescence immunoassay combined with the magnetic particles (MPCLIA)	Liquid-partitioning	0.1–100 ng mL^−1^	85.67–108.67%	0.05 ng mL^−1^	[197]
Corn germ oil Peanut oil	An immunoassay based on both recombinant antibody and its mimotope	Liquid-Liquid extraction	0.242.21 ng mL^−1^	86.70–116.20%	0.13 ng mL^−1^	[206]
Vegetable oil	Immobilized immunosensor based on the hybrid gold nanoparticles-poly 4-aminobenzoic acid supported graphene	-	0.01–25 ng mL^−1^	-	0.001 ng mL^−1^	[207]
Peanut oil	UCNPs-BPNSs aptamer	-	0.2–500 ng mL^−1^	92.89–99.24%	0.028 ng mL^−1^	[224]
Peanut oil	Dual-terminal stemmed aptamer beacon, aggregation-induced emission	Liquid-Liquid Extraction	40–300 ng mL^−1^	93.59–109.30%	27.3 ng mL^−1^	[225]
Corn oil	A chimeric aptamer-based gold nanoparticles aptasensor	-	5–5120 nM	91.50–117.60%	1.88 nM	[226]
Corn oil Peanut oil	An electrochemical aptasensor base on an AuNPs/ZIF-8 nanocomposite	-	10.0–1.0 × 10^5^ pg mL^−1^	93.49–106.90%	1.82 pg mL^−1^	[227]
Peanut oil	An electrochemical aptasensor base on an AuNPs/Zn/Ni-ZIF-8–800@ graphene nanocomposite	-	0.18–100 ng mL^−1^	80.26–109.60%	0.18 ng mL^−1^	[228]
Oil	An aptasensor of hybridization chain reaction and Zn^2+^-dependent DNAzyme catalyzed cleavage	-	0.4–16 nmol L^−1^	92.20–107.80%	0.22 nmol L^−1^	[237]
Oil	Fabricating electrochemical aptasensors	-	0.04–0.10 ng m L-1	94.5–103.3%	0.002 ng m L^−1^	[184]
Peanut oil	Electrochemical immunosensor base on AFB_1_-BSA-QDs	-	0.08–80 μg kg^−1^	102.70–113.30%	0.05 μg kg^−1^	[185]
Peanut oil	SERS aptasensor	-	0.0001–100 ng·mL^−1^	96.60–115.00%	0.40 pg·mL^−1^	[174]
Peanut oil	SERS aptasensor	-	0.001–10 ng mL^−1^	94.70–109.00%	0.54 pg mL^−1^	[169]
Peanut oil	Q-dots-aptamer-GO fluorescence quenching system	-	1.6–160 μM	-	1.4 nM	[242]
Peanut oil	Atomic absorption spectroscopy	-	2.5–240 μg kg^−1^	-	0.04 μg kg^−1^	[241]
Peanut oil	SERS aptasensor	–	0.01–100 ng mL^−1^	91.09–105.73%	0.0036 ng mL^−1^	[170]
Peanut oil	SERS aptasensor with NH_2_-Rh-Au@Ag CSNPs	Solid Phase Extraction	0.1–5.0 ng mL^−1^	-	0.03 ng mL^−1^	[244]
Olive oil Peanut oil	Dual DNA tweezers nanomachine	-	0.08–10 ppb	90.00–110.00%	0.035 ppb	[245]
Peanut oil	Electrochemical aptasensor based on smart host-guest recognition of β-cyclodextrin polymer	-	0.1 × 10^−4^–10 ng mL^−1^	94.50–106.70%	0.049 pg mL^−1^	[246]
Peanut oil	A dual signal amplified aptasensor based on DNA walker, (DTNs) and network (HCR)	-	1−1000 pg mL^−1^	87.56–105.28%	0.492 pg mL^−1^	[236]
Corn oil	A novel fluorescence aptasensor based on mesoporous silica nanoparticles	-	0.5–50 ng mL^−1^	90.30–92.40%	0.13 ng mL^−1^	[262]
Peanut oil	Dual-terminal proximity aptamer probes	-	1.0–200 ng mL^−1^	90.30–102.91%	0.9 ng mL^−1^	[263]
Sesame oil Olive oil Peanut oil Soybean oil	An aptamer-based MCE-LIF	-	0.05–5.0 ng mL^−1^	95.29–109.19%	0.026 ng mL^−1^	[264]
Peanut oil	A simple fluorescent AFB_1_ sensor based on a humic acid/carbon dots system	-	0.1–0.8 ng mL^−1^	103.80–108.00%	70 pg mL^−1^	[171]
Peanut oil	SERS aptasensor	-	0.01–100 ng mL^−1^	90.40–113.10%	5.0 ng mL^−1^	[172]
Edible oil	Immunoaffinity chromatography fluorometer	Immunoaffinity column clean-up	1.0–32.2 μg kg^−1^	-	1 μg kg^−1^	[265]
Corn oil	An MIP-ECP-ECL sensing platform based on CH_3_NH_3_PbBr_3_ quantum dots (MAPB QDs)@SiO_2_	-	10^−5^–10 ng mL^−1^	102.00–110.00%	8.5 fg mL^−1^	[173]
Soybean oil	Terahertz spectroscopy (photoelectric techniques)	-	-	-	2 μg kg^−1^	[162]
Peanut oil Corn oil	ELC based on Escherichia coli	-	0.01–0.3 μg mL^−1^	90.00–112.00%	1 μg mL^−1^	[266]

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
