# Peer review of "Recent Progress on Techniques in the Detection of Aflatoxin B1 in Edible Oil: A Mini Review"

_molecules, 2022, doi:10.3390/molecules27196141_

Round 1
Reviewer 1 Report
Author have written Recent progress on techniques in the detection of aflatoxin B1 2 in edible oil : a mini review. It is 47 page long. It is very long review. It needs to tide up.
It is great that authors introduce mycotoxin types in figure 1. please move figure 1 before line 85. The figure needs to appear after mentioned. not after two pages later.
The figure 2 should be moved to line 90. same comment as previous.
There is missing parts
First one method how to collect paper from which year, which key words, how many papers have been collected.
Second analysis of papers.
There are lot of important information in the a long table from page 20 to 29. The table should be after the method. The table should be analysis as result.
please put limited of detections of each detection method in a graph.
Please discuss which method is lowest detection and based on the results.
The section of 5. Electrochemical biosensing for AFB1 detection in edible oil is still belong to method for detection. please move as 4.4.
THe section 6 6. Bioinspired recognition elements for AFB1 biosensors in edible oil is still belong to method for detection. please move as 4.5.
Author Response
Please see the attachment.
TRANSLATE with x English| Arabic | Hebrew | Polish |
| Bulgarian | Hindi | Portuguese |
| Catalan | Hmong Daw | Romanian |
| Chinese Simplified | Hungarian | Russian |
| Chinese Traditional | Indonesian | Slovak |
| Czech | Italian | Slovenian |
| Danish | Japanese | Spanish |
| Dutch | Klingon | Swedish |
| English | Korean | Thai |
| Estonian | Latvian | Turkish |
| Finnish | Lithuanian | Ukrainian |
| French | Malay | Urdu |
| German | Maltese | Vietnamese |
| Greek | Norwegian | Welsh |
| Haitian Creole | Persian |

Reviewer 2 Report
The authors review the various techniques available to detect aflatoxins in edible oils.
The paper is written quite comprehensively, but I have some suggestions for improvements:
- Commercially available techniques should be better distinguished from research techniques (single studies)
- Some rapid tests are commercially available based on ELISA or antigen tests (similar to COVID-19 tests).
Some detailed comments:
Line 12: I would disagree that it is frequent and inevitable. In Europe, at least, the exceedance of aflatoxin levels is rather seldom
Line 13: define all abbreviations at first use
Line 14: most of the effects stated are acute toxic effects occurring at higher levels, but not the trace contamination levels found in foods. Here, the carcinogenic and chronic toxic properties are of most relevance.
Line 19: perhaps “advantages and weaknesses” are better words than merits and demerits
Line 42: I cannot believe the number of 42% animal deaths
Line 86: do you mean reproductive instead of reproducible?
Table 1: I believe there are limits in the EU for maize and other cereals, which are intended for processing as foods. Hence, it is not allowed to process contaminated grains into oils.
Section 6.2: the text is quite long without any subheading. This is somehow tiresome to read.
Table 2: improve formatting, e.g. increase width of second column
Author Response

(The authors gave the same response as above.)

Reviewer 3 Report
The manuscript “Recent progress on techniques in the detection of aflatoxin B1 in edible oil : a mini review” should be improved, in particular, English editing is recommended to enhance the overall manuscript presentation.
A literature survey revealed that there are quite a number of “review” published which summarised the detection of mycotoxins in edible oils using various methods/techniques. What is the significance of the present review?
Instead of a Conclusion, authors should provide a critical perspectives regarding the topic reviewed. Each point should be discussed and elaborated.
What are the major challenges and solutions to the topic reviewed?
All these points should be included in the revised manuscript.
Please make sure the scientific names of the microorganisms and plants are in italic.
Reference style is not consistent. Eg. some journals ‘names were abbreviated, some were not.
Author Response

(The authors gave the same response as above.)

Reviewer 4 Report
In the article entitled: “Recent progress on techniques in the detection of aflatoxin B1 in
edible oil: a mini review” authors describes different techniques in the detection of aflatoxin B1.
This is an interesting article. Written by peoples with a very high level of knowledge of the topic. This article has been written in a compact manner. From the structure point of view, the sub-chapter are appropriate to the adopted objective of the review work. The methods described may be useful for the industry and researchers.
Title
The title and the aim of the study are clearly constructed.
Abstract
The abstract includes the aim of the study, the technique described and contain the principal results and conclusions.
The mini-review
The review describes the importance of the topic and states the problem being investigated. Authors correctly interpreted and described the significance of their topic. A lot of techniques for aflatoxin B1 are described. Were also present the advantages and disadvantages of using them in practice. They skillfully referred to the results of other researchers. Literature references are the most current (usually from the last 2-3 years).
Author Response
Thank you very much for your affirmation of this manuscript!
TRANSLATE with x English| Arabic | Hebrew | Polish |
| Bulgarian | Hindi | Portuguese |
| Catalan | Hmong Daw | Romanian |
| Chinese Simplified | Hungarian | Russian |
| Chinese Traditional | Indonesian | Slovak |
| Czech | Italian | Slovenian |
| Danish | Japanese | Spanish |
| Dutch | Klingon | Swedish |
| English | Korean | Thai |
| Estonian | Latvian | Turkish |
| Finnish | Lithuanian | Ukrainian |
| French | Malay | Urdu |
| German | Maltese | Vietnamese |
| Greek | Norwegian | Welsh |
| Haitian Creole | Persian |
Round 2
Reviewer 1 Report
The authors have been requested to arrange a figure regarding the table 2 to illustrate which type of method is sensitive which one is second which one is less sensitive see https://doi.org/10.3390/applnano3020006 . Authors must provide the figure include two graphs.
Please provide the graph 1 use y axle with limit of detection from column 6 (LOD) and x axle with average of each methods such as Chromatographic technology, Spectroscopic technology, Immunological technology, Nanomaterial-based biosensors, Bioinspired recognition elements for AFB1 biosensors.
Please arrange graph 2
use y axle with limit of detection and x axle with average of each methods:
HPLC, TLC, Fluorescence spectrophotometry, Infrared (IR) spectroscopy, Terahertz (THz) spectroscopy, ELISA, Electrochemical biosensing, Amperometric biosensors, Aptamers, SERS, Molecularly Imprinted Polymers,
Line 98-100
'We only included the detection technology targeted at aflatoxin B1 in edible oil. We do not included other types of toxins'
Please use third person passive tense
Line 661 SER should have section number. It should be Italic.
Table 2 reference [236]
'A dual signal amplified aptasensor based on DNA walker, (DTNs) and network (HCR)' should be smaller font as others
same comments as previous
for reference [242] [241] [244] in Table 2
Author Response
Dear reviewer:
We are so grateful for your question and kind recommendation. We have taken your advice and references to make Figure 4.
According to your requirements, We have changed the syntax and optimized the table.
And please see the attachment.
Round 3
Reviewer 1 Report
It is great efforts authors have made a figure 4. Can you please describe which technical are among the lowest LOD in a and b.
Please use a. b instead of A. B for the figure 4.